# Small Molecular Weight Aldose (d-Glucose) and Basic Amino Acids (l-Lysine, l-Arginine) Increase the Occurrence of PAHs in Grilled Pork Sausages

**DOI:** 10.3390/molecules23123377

**Published:** 2018-12-19

**Authors:** Wen Nie, Ke-zhou Cai, Yu-zhu Li, Shuo Zhang, Yu Wang, Jie Guo, Cong-gui Chen, Bao-cai Xu

**Affiliations:** 1School of Food Science and Engineering, Hefei University of Technology, Hefei 230009, Anhui Province, China; 17356511419@163.com (W.N.); 1932526946@163.com (Y.L.); 18856023156@163.com (S.Z.); wyll_ah92@163.com (Y.W.); hfutkaylee@163.com (J.G.); chencg1629@hfut.edu.cn (C.C.); baocaixu@163.com (B.X.); 2Key Laboratory on Deep Processing of Agricultural Products for Anhui Province, Hefei 230009, Anhui Province, China; 3Engineering Research Centre of Bio-Process, Ministry of Education, Hefei 230009, Anhui Province, China

**Keywords:** amino acids, carbohydrates, acidity, polarity, molecular weight

## Abstract

(1) Background: Amino acids and carbohydrates are widely used as additives in the food industry. These compounds have been proven to be an influencing factor in the production of chemical carcinogenic compounds polycyclic aromatic hydrocarbons (PAHs). However, the effect of the properties of the amino acids and carbohydrates on the production of PAHs is still little known. (2) Methods: We added different (i) R groups (the R group represents an aldehyde group in a glucose molecule or a ketone group in a fructose molecule); (ii) molecular weight carbohydrates; (iii) polarities, and (iv) acid-base amino acids to pork sausages. The effects of the molecular properties of carbohydrates and amino acids on the formation of PAHs in grilled pork sausages were investigated. (3) Results: The results showed that a grilled sausage with aldehyde-based d-glucose was capable of producing more PAHs than a sausage with keto-based d-fructose. A higher PAH content was determined in the grilled pork sausage when the smaller molecular weight, d-glucose, was added compared with the sausage where the larger molecular weight, 4-(α-d-glucosido)-d-glucose and cellulose were added. The addition of basic amino acids (l-lysine, l-arginine) was capable of producing more PAHs compared with the addition of acidic amino acids (l-glutamic acid, l-aspartate). When amino acid containing a benzene ring was added, a smaller volume of PAHs was produced compared with the addition of other amino acids. (4) Conclusions: Our study suggests that systematic consideration of molecule properties is necessary when using food additives (amino acids and carbohydrates) for food processing.

## 1. Introduction

Polycyclic aromatic hydrocarbons (PAHs), formed through incomplete combustion of wood or gasoline, are regarded as potentially genotoxic and carcinogenic to humans [1,2]. The occurrence of PAHs in food, such as edible oils, meat, and dairy products, has been regarded as a consequence of high-temperature processing [3]. Previous epidemiological studies have speculated that food consumption might contribute 88–98% of PAHs exposure. This is especially the case in populations that are non-smoking and not subject to occupational exposure [4]. Among the sources of PAHs exposure, meat products account for the largest proportion. The formation and control of PAHs during the processing of meat products is topical in current research.

The definite mechanism for the formation of PAHs is not well understood. Some researchers proposed that they might be formed through free radical reactions, intramolecular addition or the polymerization of small molecules [5,6,7]. In meat products, PAHs are formed during processing at a high temperature, such as by smoking, drying, roasting, and grilling [8,9,10]. Grilled sausages are a common food and can be contaminated with high concentrations of PAHs [11]. Amino acids, a kind of food additive, are often used to improve the sensory properties (color, taste, and texture) of meat products to meet the needs of consumers [12,13]. Some scholars have studied the pyrolysis of aliphatic α-amino acids, glutamine, glutamic acid, and aspartic acid and the formation of PAHs. The results show that single chemical components of glutamine, glutamic acid, and aspartic acid can pyrolyze to form PAHs at high temperatures. The author speculates that all three acids appear to break down to smaller “building blocks” and then pyrosynthesize the PAHs [14]. Other scholars have explored whether glucose can enhance the pyrolysis of proline to form PAHs and found that glucose can provide a low-temperature pathway for the decomposition of proline, thus, enhancing the pyrolysis of proline to form PAHs [15]. The above studies indicate that some amino acids can be pyrolyzed to form PAHs in a single simulated system, and glucose can promote the pyrolysis of proline to form PAHs. However, meat products are a complex matrix. The formation of PAHs in meat products by glucose and amino acids is significantly different from the formation of PAHs by the pyrolysis of glucose and amino acids in a single chemical system. There is still no information on the effect of amino acid types on the PAHs formation in cooked meat.

In addition to nutritional properties, carbohydrates can also be used as a sweetener, gel, thickener, or stabilizer in the meat industry. The physical and chemical properties, the degree of polymerization, solubility, viscosity, film formation, and gelation greatly influence the retention and release of flavor [16,17]. In addition, carbohydrates retain aromatic components and reduce flavor loss during meat processing [17]. Other studies considered using polysaccharide to promote water holding capacity and texture [18]. Although many carbohydrates have been successfully used to increase the flavor, color, and texture of meat, there is still limited research on their safety including the effect of PAHs production in cooked meat.

Therefore, it is of interest to specifically study the effects of amino acids and carbohydrate types on PAHs formation in cooked meat. In this paper, we tested the effects of carbohydrates with a different R-base (an aldehyde group in a glucose molecule and a ketone group in a fructose molecule) and different molecular weights, and the property of amino acids (with different polarities and a different acid-base) on the formation of twelve kinds of PAHs in grilled sausages. All of these PAHs are noted as potential food contaminants by the United States Environmental Protection Agency [19].

## 2. Results and Discussion

### 2.1. Validation

The detection limits, recoveries, and precision for twelve kinds of PAHs data are presented in Table 1. Linearity was determined using external standard plot method. The HPLC of all 12 PAHs in the master sample is presented in Figure 1. The high R^2^ values indicated good linearity over the concentration range. As can be seen, the limit of detection (LOD) and limit of quantification (LOQ) of all PAHs ranged from 0.03 to 0.18 μg kg^−1^ and 0.10 to 0.91 μg kg^−1^ and met the Commission of the European Communities (2011b). Relative standard deviations (RSD) and recovery in our study ranged from 1.23 to 9.71% and 79.22 to 105.78%, respectively. According to the regulation [20], recovery sets should be in the range of 50–120%; recovery in the experiments varied between 79.22% and 105.78%, which is in good accordance with the regulation. The precision was adequate, with RSD < 10%. It has been well established that the presence of some impurities in meat samples, such as aliphatic hydrocarbons, fatty acids, phenols, and polycyclic organic compounds, may greatly reduce the extraction efficiency of PAHs [21].

### 2.2. Effect of Carbohydrate Characteristics on PAHs Content

Carbohydrate, as an important nutrient of meat and a common additive in meat products, was reported to have a notable influence on PAHs production during meat processing [22,23]. In this study, the effects of aldose (d-glucose) and ketose (d-fructose) on PAHs production during meat processing were investigated by adding d-glucose and d-fructose to grilled pork sausages. The d-glucose, 4-(α-d-Glucosido)-d-glucose, and cellulose were used to explore the effect of carbohydrate molecular weight on the formation of PAHs during meat processing. The results showed there was no significant difference in the content of PAHs between the grilled pork sausage with d-fructose and the blank control sausage (*p* > 0. 05). However, compared with these two groups, the occurrence of PAHs in the grilled pork sausages with d-glucose was significantly increased (*p* < 0.05) (details are shown in Table 2). The reason for this phenomenon might be due to d-glucose containing free aldehyde groups, which can be decomposed to smaller molecular aldehyde compounds at high temperatures, and these aldehydes can be further subjected to complex cracking, polymerization or condensation reaction to ultimately produce PAHs. However, the ketone group in d-fructose needs to be converted into an aldehyde group at a higher temperature to further generate PAHs by a cleavage reaction [24].

When comparing the influence of the carbohydrate’s molecular weight on the content of PAHs in grilled pork sausages, it was found that the concentration of PAHs in sausages supplemented with d-glucose or 4-(α-d-Glucosido)-d-glucose was significantly higher than that of the blank control group. However, the content of PAHs in the grilled sausage to which cellulose had been added was not significantly different from that of the control group. In addition, the PAHs content in the grilled sausage with d-glucose was significantly higher than that of in the grilled pork sausage with 4-(α-d-Glucosido)-d-glucose (*p* < 0.05) (details are shown in Table 2). A higher PAHs content was determined in the grilled sausage with smaller molecular weight, d-glucose, when compared with the larger molecular weight, 4-(α-d-Glucosido)-d-glucose and cellulose. However, there was no significant difference in the content of PAHs in grilled pork sausages with the addition of small molecular weight d-fructose and large molecular weight cellulose (*p* > 0.05). This result may be due to the effect of carbohydrates on PAHs production being affected by both the R group and molecular weight, and the effect of R group on PAHs formation was greater than the effect of molecular weight. Some scholars explored the formation of PAHs from the pyrolysis of carbohydrates, amino acids, and fatty acids, and it was found that high-temperature pyrolysis of d-glucose produced more PAHs than starch, which result is consistent with our research [25]. Other scholars studied the mechanism of dehydration, carbonization, decarbonylation, decarboxylation, dehydrogenation, and cross-linking of cellulose at high temperatures when studying the mechanism of pyrolysis of cellulose to form PAHs. It was proposed that PAHs could be formed through transformation and rearrangement [26]. The study also found that the yield of PAHs produced by pyrolysis of d-glucose and sucrose was higher than that of cellulose. This result is consistent with our findings. However, the current mechanism of the molecular weight of carbohydrates on the formation of PAHs during high-temperature processing is still unclear. 

### 2.3. Effect of Amino Acid Properties on PAHs Content

To explore the effects of amino acid polarity and acidity/alkalinity on the occurrence of PAHs in meat processing under high temperature, several different polar amino acids, non-polar amino acids, acidic amino acids, and basic amino acids were chosen to be added to grilled pork sausages. When investigating the effects of three non-polar amino acids on the content of PAHs in grilled pork sausages, it was found that the content of PAHs in sausages supplemented with l-proline was significantly increased compared with the blank control group (*p* < 0.05). However, the content of PAHs in grilled pork sausages added l-tryptophan or l-phenylalanine was not significantly different from that of the blank control group (*p* > 0.05) (details are shown in Table 3). The effects of three polar amino acids on the content of PAHs in grilled pork sausages were investigated. The result showed that the addition of l-threonine or l-serine significantly increased the PAHs content in grilled pork sausages when compared with the blank control group (*p* < 0.05). However, there was no significant difference between the sausage supplemented with l-tyrosine and the blank control group (*p* > 0.05) (details are shown in Table 3). The authors did not find obvious rules when comparing the effects of polar and non-polar amino acids on the formation of PAHs in grilled pork sausages. This indicates that the formation of PAHs in grilled pork sausages is not affected by the polarity of the amino acids.

When investigating the effects of two basic amino acids on the content of PAHs in grilled pork sausage, it was found that the content of PAHs in the sausage with l-lysine and l-arginine was significantly higher than that in the blank control group. However, there was no significant difference in the content of PAHs in the grilled pork sausage with l-arginine or l-lysine (*p* > 0.05). When investigating the effects of two acidic amino acids on the content of PAHs in grilled pork sausage, it was found that the content of PAHs in the sausage to which l-glutamic acid and l-aspartic acid were added was significantly higher than that in the blank control group (*p* < 0.05). However, there was no significant difference in the PAHs content between the sausages supplemented with l-glutamic acid or l-aspartic acid (*p* > 0.05). In addition, when comparing the effects of acidic amino acids (l-lysine and l-arginine) and basic amino acids (l-glutamic acid and l-aspartate) on the formation of PAHs in grilled pork sausages, the authors found that the content of PAHs in sausages with basic amino acids was significantly higher than that in grilled pork sausages with acidic amino acids (*p* < 0.05) (details are shown in Table 3). The reason may be due to the basic amino acids promoting the Maillard reaction during the grilling of the sausages. Previous studies have shown that the Maillard reaction was increased with the increase of Ph [23], and the Maillard reaction has a great relationship with the formation of PAHs [23,27]. The pyrolysis of proteins produces free amino acids and these amino acids can react with reducing sugars (such as glucose) to form the Amalido compound (1-amino-1-deoxy-2-ketosaccharide) [14]. These compounds then undergo pyrolysis to produce PAHs. But the detailed relationship between Maillard reaction and PAHs production is still unclear.

### 2.4. Effect of Benzene Ring in Amino Acid on PAHs Content

The authors found a very interesting phenomenon when investigating the effects of three non-polar amino acids and three polar amino acids on the content of PAHs in grilled pork sausages. When investigating the effects of three non-polar amino acids on the content of PAHs in grilled pork sausages, it was found that the content of PAHs in sausages supplemented with l-tryptophan and l-phenylalanine was significantly lower than proline (as shown in Figure 2A). When investigating the effects of three polar amino acids on the content of PAHs in grilled pork sausages, the results showed that the content of PAHs in sausages supplemented with l-tyrosine was significantly lower than the sausage with l-serine or l-threonine (*p* < 0.05), but there was no significant difference compared with the control group (*p* > 0.05) (as shown in Figure 2B). Aspartic acid and proline degrade at high temperatures to produce PAHs, such as phenanthrene and aromatics. Tryptophan could not be degraded to produce PAHs when treated at high temperature [15]. This is consistent with the results of our present experiment. It is well known that l-tryptophan, l-phenylalanine, and l-tyrosine are all aromatic amino acids, and each contains a benzene ring. Therefore, it is reasonable to speculate that the benzene ring in the amino acid molecule is a significant reason that these aromatic amino acids hardly influence the production of PAHs. This may be due to the benzene ring in the amino acid molecule being structurally stable and not easily cleaved into small molecular substances, for example, an aldehyde, an enyne, etc. at a high temperature.

## 3. Materials and Methods

### 3.1. Standards and Reagents

d-glucose (CAS number: 50-99-7), d-fructose (CAS number: 53188-23-1), 4-(α-d-Glucosido)-d-glucose (CAS number: 6363-53-7), cellulose (CAS number: 9004-34-6), Non-polar amino acids: l-proline (CAS number: 147-85-3), l-tryptophan (CAS number: 73-22-3) and l-phenylalanine (CAS number: 63-91-2 ); Polar amino acids: l-tyrosine (CAS number: 60-18-4), l-threonine (CAS number: 72-19-5) and l-serine (CAS number: 56-45-1); Basic amino acids: l-Lysine (CAS number: 56-87-1) and l-arginine (CAS number: 74-79-3); Acidic amino acids: l-glutamic acid (CAS number: 56-86-0) and l-aspartic acid (CAS No.: 56-84-8) (See Table 4 for specific information on the compounds studied above). The above reagents were all more than 99% pure and were purchased from Aladdin Reagent (Shanghai Aladdin Biochemical Technology Co., Ltd., Shanghai, China). Cyclohexane, dichloromethane (analytical pure) and acetonitrile (chromatographic pure) were purchased from Sinopharm Group. Twelve kinds of PAHs that were used as standards (Naphthalene (NA), Acenaphthene (Ac), Fluorene (FLU), Fluoranthene (FLT), Benzo[a]anthracene (BaA), Chrysene (CHR), Benzo[b]fluoranthene (BbFA), Benzo[k]fluoranthene (BkFA), Benzo[a]pyrene (BaP), Dibenzo[a,h]anthracene (DBahA), Benzo[g,h,i]perylene (BPE), and Indeno[1,2,3-c,d]pyrene (IPY) ) were purchased at a purity of more than 99% from Toronto Research Chemicals, Brisbane, North York, ON, Canada.

### 3.2. Sample Preparation

Fresh hind legs and fat from pigs were purchased from a Carrefour supermarket in Hefei, China. After the visible connective tissues and fat had been removed, the meat was cut into small pieces and ground twice using an electric meat mincer (sieve plate aperture 4 mm; TJ12-H, Henglian, Guangdong, China). The fat was also ground twice using an electric meat mincer with a sieve plate aperture of 6 mm. Four thousand grams of ground meat, 1000 g of fat, 100 g of sodium chloride were thoroughly mixed, then divided into 14 parts, each about 360 g. One point eight grams of an amino acid or carbohydrate was added to each portion and then mixed well. After curing for 6 h at 6–10 °C, the two groups were stuffed into edible casings (Shengguan Casing Company, Guangxi, China) to form sausages. The diameter, length and weight of the sausages were 1.9 cm, 20 cm, and 100 ± 10 g, respectively. All experiments were carried out in triplicate.

### 3.3. Grilling Tools and Cooking Procedures

In this study, the sausages were grilled using an electric oven (Electrolux, EOT5004K, Guang dong, China). The sausages were placed on a metal grid 12 cm below the heating source and baked for 20 min at 240 °C and flipped every 5 min.

### 3.4. Extraction and Clean-Up

The extraction and clean-up procedures were performed as described by Farhadian et al. [28]. After completion of the heat treatment, the sausage was removed and cooled to room temperature. Five grams of minced sausage was precisely weighed and placed into a 50 mL centrifuge tube containing 25 mL of cyclohexane. After shaking for 5 min with a quick mixer (SK-1, Jintan is Jairel Electric Co., Ltd., China), the centrifuge tube was subjected to ultrasonic treatment for 20 min (SB-5200D; Xinzhi Biotechnology Co., Ltd., Ningbo, China). After sonication, the solution was applied to a Florisil solid phase extraction (SPE) column previously treated with dichloromethane (3 mL) and cyclohexane (5 mL). The PAHs adsorbed by the SPE column were eluted with 9 mL of cyclohexane dichloromethane (3:1 *v*/*v*). The eluate was evaporated by rotary evaporation (40 °C, 30 rmp) until only 1–2 mL of concentrate remained. The concentrate was transferred into a 10 mL centrifuge tube and dried in a nitrogen atmosphere. A total of 2 mL of acetonitrile was then added to dissolve the extract, which was then filtered through a 0.22 µm membrane for analysis by HPLC.

### 3.5. HPLC Analysis of PAHs

The PAH analyses were performed based on the method described by Farhadian et al. [28]. PAH analysis was carried out using an HPLC apparatus (Agilent, Santa Clara, CA, USA) equipped with a 600 controller pump, fluorescence detector (G1312, Agilent, USA) and a 20 μL loop injector. A PAH column (250 mm × 4.6 mm, 5 μm particle size) (Agilent, Santa Clara, CA, USA) was used. The mobile phase was constituted of acetonitrile and water. The elution conditions applied were: 0–3 min, 60% of acetonitrile isocratic; 3–15 min, 60–100% of acetonitrile; 15–46 min, 100% of acetonitrile isocratic; 46–53, 100–60% of acetonitrile, gradient. The flow rate was 1.0 mL min^−1^. A fluorescence detector operated at excitation/emission wavelength 265/327 nm for NAP, 285/320 nm for Ac, 256/300 for FLU, 275/450 nm for FLT, 274/382 nm for BaA, 260/360 for CHR, 283/430 for BbFA and BkFA, 285/410 for BaP, 284/395 for DBahA, 290/410 for BPE, and 301/480 for IPY. Separation was performed under isocratic conditions. Each PAH sample solution was passed through a 0.22 μm filter before injection into the HPLC system. The quantification of PAHs was performed using an external calibration curve method. The quantification of 12 PAHs was carried out through the external standard method.

### 3.6. Method Validation

The calibration curve, linearity (tested through square correlation coefficients R^2^) and limits of detection and quantification (LOD and LOQ) were obtained using the signal-tonoise ratio of S/N = 3 and S/N = 10, respectively [29]. Accuracy was evaluated using spiked sausage samples at four levels of concentration (ranging from 0.25 to 20.0 μg kg^−1^), and recoveries were calculated. The recoveries were calculated from the differences in total amounts of each PAHs between the spiked and unspiked samples. For precision, the sample was spiked with two levels (ranging from 0.25 to 8.0 μg kg^−1^) on two different days, by a single analyst using the same equipment, and the relative standard deviation (RSD) was determined.

### 3.7. Statistical Analysis

The sample data are provided as the mean ± standard deviation. The data were subjected to analysis of variance (ANOVA) and Duncan’s multiple range test for statistical significance (*p* < 0.05) using the SPSS 17.0 software system (SPSS Inc., Chicago, IL, USA).

## 4. Conclusions

This study tested the effects of molecule properties of carbohydrates and amino acids on PAHs formation in grilled pork sausage. The results showed that addition of aldehyde-based d-glucose could significantly increase the PAHs production when compared with that of the keto-based d-fructose group. The grilled sausage with smaller molecular weight, d-glucose, was determined to have more PAHs content as compared with the sausage with larger molecular weight, 4-(α-d-Glucosido)-d-glucose and cellulose. The addition of basic amino acids (l-lysine, l-arginine) was capable of producing more PAHs in grilled sausages than adding acidic amino acids (l-glutamic, l-aspartate). The addition of the amino acid containing a benzene ring produced fewer PAHs than other amino acids in the grilled sausages. Our study suggests that systematic consideration of the molecule properties is necessary when using food additives (amino acids and carbohydrates) for food processing.

## Figures and Tables

**Figure 1 molecules-23-03377-f001:**
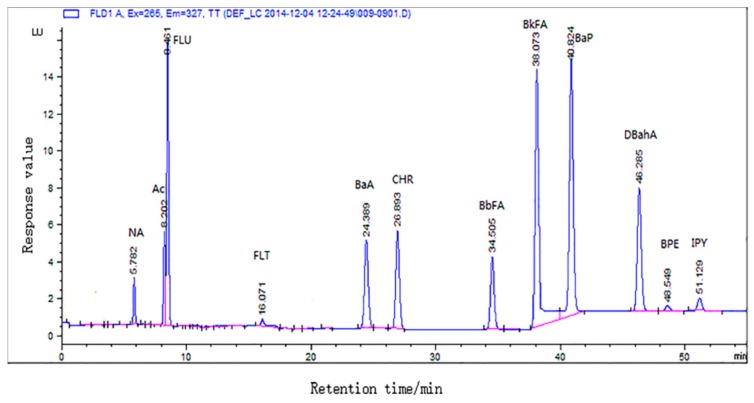
Standard chromatogram of 12 polycyclic aromatic hydrocarbons.

**Figure 2 molecules-23-03377-f002:**
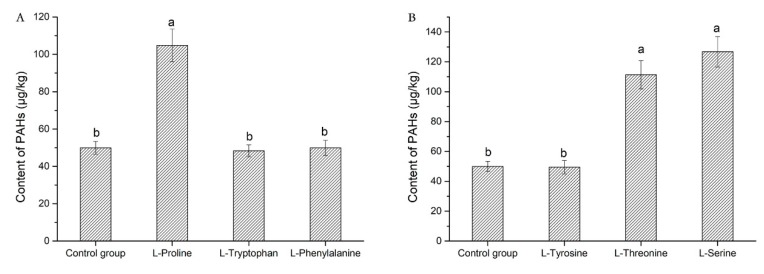
Effect of benzene ring in amino acid on polycyclic aromatic hydrocarbons (PAHs) content. (**A**) The effects of three non-polar amino acids on the content of PAHs in grilled pork sausages; (**B**) The effects of three polar amino acids on the content of PAHs in grilled pork sausages; a, b, different letters indicate significant differences in PAHs content between any two bar graphs in Figure 2A,B (*p* < 0.05).

**Table 1 molecules-23-03377-t001:** Parameters of validation of 12 polycyclic aromatic hydrocarbons (PAHs) in sausages.

PAHs	Linear Range (ng mL^−1^)	Correlation Coefficients (R^2^)	LOD (μg kg^−1^)	LOQ (μg kg^−1^)	Recovery ^a^ (%)	RSD ^b^ (%)
NA	0.10–10.00	0.9999	0.05	0.17	105.78	1.23
AC	0.10–10.00	0.9996	0.18	0.60	93.46	1.46
FLU	0.20–10.00	0.9999	0.06	0.20	84.54	5.54
FLT	0.20–10.00	0.9993	0.16	0.53	83.83	7.89
BaA	0.25–5.00	0.9998	0.09	0.30	90.52	4.37
CHR	0.30–7.50	0.9999	0.10	0.91	92.56	3.42
BbFA	1.00–20.00	0.9995	0.06	0.20	96.74	1.90
BkFA	1.00–20.00	0.9999	0.08	0.27	95.33	2.05
BaP	0.20–10.00	0.9999	0.11	0.37	91.31	4.55
DBahA	0.10–10.00	0.9991	0.03	0.10	84.53	9.71
BPE	0.20–10.00	0.9998	0.04	0.20	88.34	4.56
IPY	0.30–10.00	0.9997	0.06	0.30	79.22	6.18

Explanations: LOD–limit of detection; LOQ–limit of quantification; RSD–relative standard deviation of repeatability; ^a^ mean recoveries of four different spiking levels in triplicate in the same day; ^b^ mean relative standard deviations (RSD) of two different spiking levels in triplicates on two different day; Naphthalene (NA), Acenaphthene (Ac), Fluorene (FLU), Fluoranthene (FLT), Benzo[a]anthracene (BaA), Chrysene (CHR), Benzo[b]fluoranthene (BbFA), Benzo[k]fluoranthene (BkFA), Benzo[a]pyrene (BaP), Dibenzo[a,h]anthracene (DBahA), Benzo[g,h,i]perylene (BPE), Indeno[1,2,3-c,d]pyrene (IPY).

**Table 2 molecules-23-03377-t002:** Concentrations (*n* = 3) of 12 PAHs produced as a consequence of adding different carbohydrates to grilled sausages.

Species	NA	Ac	FLU	FLT	BaA	CHR	BbFA	BkFA	DBahA	BPE	BaP	IPY	∑PAH_12_
Control group	17.93 ± 1.35 c	12.99 ± 1.02 c	2.79 ± 0.14 c	ND	2.21 ± 0.16 c	3.62 ± 0.10b c	ND	7.52 ± 0.36 b	0.71 ± 0.05 c	1.25 ± 0.08 c	0.88 ± 0.12 d	ND	49.90 ± 3.46 c
d-Fructose	18.36 ± 1.41 c	11.15 ± 1.14 c	3.13 ± 0.26 c	1.06 ± 0.15 c	3.04 ± 0.27 bc	2.47 ± 0.16 d	ND	6.01 ± 0.33 c	1.04 ± 0.11 b	1.37 ± 0.09 b	1.26 ± 0.06 c	ND	48.89 ± 4.05 c
d-Glucose	34.25 ± 1.22 a	23.45 ± 1.68 a	7.66 ± 0.23 a	4.24 ± 0.18 a	7.41 ± 0.36 a	6.75 ± 0.36 a	0.47 ± 0.03 a	13.76 ± 1.28 a	3.16 ± 0.30 a	3.07 ± 0.26 a	5.59 ± 0.22 a	ND	109.81 ± 6.49 a
4-(α-d-Glucosido)-d-glucose	25.17 ± 2.23 b	17.42 ± 1.35 b	5.06 ± 0.44 b	2.31 ± 0.18 b	4.01 ± 0.23 b	4.48 ± 0.24 b	0.29 ± 0.01 b	8.47 ± 0.31 b	1.35 ± 0.10 b	1.54 ± 0.10 b	3.46 ± 0.25 b	ND	73.56 ± 5.67 b
Cellulose	17.16 ± 1.27 c	11.56 ± 1.06 c	3.03 ± 0.21 c	ND	2.44 ± 0.17 c	3.53 ± 0.28 c	ND	6.46 ± 0.31 c	0.82 ± 0.03 bc	1.39 ± 0.11 b	1.12 ± 0.18 cd	ND	47.51 ± 3.38 c

Explanations: the results are expressed as the mean and standard deviation; ND—Means no PAHs detected; a, b, c, d, different letters indicate significant differences in PAHs content between any two rows in Table 2 (*p* < 0.05); Naphthalene (NA), Acenaphthene (Ac), Fluorene (FLU), Fluoranthene (FLT), Benzo[a]anthracene (BaA), Chrysene (CHR), Benzo[b]fluoranthene (BbFA), Benzo[k]fluoranthene (BkFA), Benzo[a]pyrene (BaP), Dibenzo[a,h]anthracene (DBahA), Benzo[g,h,i]perylene (BPE), Indeno[1,2,3-c,d]pyrene (IPY).

**Table 3 molecules-23-03377-t003:** Concentrations (*n* = 3) of 12 PAHs produced as a consequence of adding different amino acids to grilled sausages.

	Species	NA	Ac	FLU	FLT	BaA	CHR	BbFA	BkFA	DBahA	BPE	BaP	IPY	∑PAH12
	Control group	17.93 ± 1.35 de	12.99 ± 1.02 cd	2.79 ± 0.14 cd	ND	2.21 ± 0.16 c	3.62 ± 0.10 c	ND	7.52 ± 0.36 c	0.71 ± 0.05 c	1.25 ± 0.08 cd	0.88 ± 0.12 b	ND	49.90 ± 3.46 d
Non-polar amino acid	l-proline	30.42 ± 2.26 c	27.10 ± 1.54 b	6.17 ± 0.33 b	3.16 ± 0.18 a	5.06 ± 0.27 ab	7.83 ± 0.24 b	1.01 ± 0.04 a	17.43 ± 1.33 a	1.64 ± 0.11 b	2.73 ± 0.13 b	2.23 ± 0.20 a	ND	104.78 ± 8.72 b
	l-tryptophan	18.35 ± 1.42 de	11.43 ± 1.24 d	3.01 ± 0.26 c	ND	1.88 ± 0.13 d	3.59 ± 0.21 c	ND	7.17 ± 0.45 c	0.75 ± 0.07 c	1.16 ± 0.11 d	1.01 ± 0.16 b	ND	48.35 ± 3.18 d
	l-phenylalanine	16.77 ± 1.37 e	13.21 ± 1.10 cd	2.85 ± 0.24 cd	ND	2.35 ± 0.17 c	3.81 ± 0.26 c	ND	8.02 ± 0.43 c	0.67 ± 0.05 c	1.36 ± 0.14 cd	0.85 ± 0.11 b	ND	49.89 ± 4.05 d
Polar amino acid	l-tyrosine	18.19 ± 1.52 de	12.15 ± 1.13 d	2.66 ± 0.22 d	ND	1.99 ± 0.14 cd	3.77 ± 0.28 c	ND	7.86 ± 0.46 c	0.69 ± 0.04 c	1.22 ± 0.11 d	0.94 ± 0.06 b	ND	49.47 ± 4.44 d
	l-threonine	39.45 ± 3.26 b	28.58 ± 2.19 ab	6.82 ± 0.31 b	ND	4.86 ± 0.20 b	7.96 ± 0.30 b	0.86 ± 0.03 a	16.54 ± 1.42a	1.56 ± 0.12 b	2.75 ± 0.21 b	1.96 ± 0.08 a	ND	111.34 ± 9.43 b
	l-serine	37.49 ± 3.33 b	34.01 ± 2.64 a	8.12 ± 0.36 a	3.72 ± 0.20 a	5.78 ± 0.26 ab	9.47 ± 0.82 ab	1.02 ± 0.01 a	19.68 ± 1.54 a	1.85 ± 0.11 b	3.27 ± 0.20 a	2.35 ± 0.17 a	ND	126.76 ± 10.17 a
Basic amino acid	l-lysine	50.48 ± 2.57 a	32.08 ± 1.88 a	8.74 ± 023 a	ND	8.76 ± 0.18 a	12.15 ± 0.32 a	ND	12.78 ± 1.14 b	3.21 ± 0.08 a	3.13 ± 0.10 a	1.52 ± 0.05 ab	ND	132.85 ± 6.13 a
	l-arginine	52.67 ± 3.36 a	30.91 ± 2.49 a	9.64 ± 0.33 a	ND	7.26 ± 0.21 a	11.61 ± 1.40 a	ND	17.89 ± 1.42 a	3.69 ± 0.33 a	3.98 ± 0.36 a	2.17 ± 0.14 a	ND	139.82 ± 8.45 a
Acidic amino acid	l-glutamic acid	20.96 ± 3.61 d	17.67 ± 2.20 c	2.37 ± 0.27 d	ND	5.45 ± 0.22 ab	7.27 ± 0.31 b	ND	10.17 ± 1.26 b	1.62 ± 0.11 b	1.86 ± 0.14 c	2.09 ± 0.10 a	ND	69.46 ± 9.27 c
	l-aspartate acid	22.76 ± 2.18 d	15.37 ± 1.06 c	3.54 ± 0.16 c	ND	3.03 ± 0.17 bc	4.59 ± 0.20 c	ND	9.54 ± 0.96 bc	2.92 ± 0.16 a	1.58 ± 0.10 c	1.14 ± 0.04 b	ND	64.47 ± 3.47 c

Explanations: the results are expressed as the mean and the standard deviation; ND—Means no PAHs detected; a, b, c, d, e, different letters indicate significant differences in PAHs content between any two rows in Table 3 (p < 0.05); Naphthalene (NA), Acenaphthene (Ac), Fluorene (FLU), Fluoranthene (FLT), Benzo[a]anthracene (BaA), Chrysene (CHR), Benzo[b]fluoranthene (BbFA), Benzo[k]fluoranthene (BkFA), Benzo[a]pyrene (BaP), Dibenzo[a,h]anthracene (DBahA), Benzo[g,h,i]perylene (BPE), Indeno[1,2,3-c,d]pyrene (IPY).

**Table 4 molecules-23-03377-t004:** Molecular weight, pKa and logP of each study compound.

	Molecular Weight (g/mol)	pKa	logP
d-Glucose	180.16	12.43 (t = 18 °C)	−3.24
d-Fructose	180.16	12.06 (t = 18 °C)	−2.23
4-(α-d-Glucosido)-d-glucose	342.30	-	−5.03
Cellulose	>50000	-	-
l-Proline	115.13	10.64 (t = 25 °C)	−2.54
l-Tryptophan	204.22	7.38 (t = 25 °C)	−1.06
l-Phenylalanine	165.19	1.24 (t = 25 °C)	−1.38
l-Tyrosine	181.19	2.20 (t = 25 °C)	−2.26
l-Threonine	119.12	5.60 (t = 25 °C)	−2.94
l-Serine	105.09	2.21 (t = 25 °C)	−3.07
l-Lysine	146.19	3.12 (t = 0 °C)	−3.05
l-Arginine	174.20	2.24 (t = 0 °C)	−4.20
l-Glutamic acid	147.13	2.23 (t = 0 °C)	−3.69
l-Aspartic acid	133.10	2.01 (t = 0 °C)	−3.89

Explanations: pKa—The acid dissociation constant refers to a specific equilibrium constant to represent the ability of an acid to dissociate hydrogen ions; logP—The logP value refers to the logarithm of the partition coefficient of a substance in n-octane (oil) and water.

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
