# Peer review of "Small Molecular Weight Aldose (d-Glucose) and Basic Amino Acids (l-Lysine, l-Arginine) Increase the Occurrence of PAHs in Grilled Pork Sausages"

_molecules, 2018, doi:10.3390/molecules23123377_

Round 1

Reviewer 1 Report

The article entitled "Small molecular weight aldose and basic amino acids increase the occurrence of PAHs in roasted pork sausages" evaluates the influence of added carbohydrates and amino acids in sausages on the formation of polycyclic aromatic hydrocarbons (PAHs). In general, it presents an interesting research topic that could have a great impact on food safety. Nevertheless, more experiments are required to support authors’ conclusions and, regrettably, this manuscript cannot be published on Molecules in its current form.

General comments:

1)     Only one keto-based carbohydrate (i.e. D-fructose) and one aldehyde-based carbohydrate (D-glucose) are evaluated. The sample size is not representative for concluding that “R-based aldehyde-based carbohydrates are capable of producing more PAHs than R-based keto-based carbohydrates during high-temperature processing of foods”. The formation of PAHs produced as a consequence of other carbohydrates of each type must be studied in order to support this conclusion.

2)     Furthermore, only three carbohydrates were selected for the evaluation of their molecular weight on the formation of PAHs during food processing. The sample size is not enough to conclude that “the smaller the molecular weight of carbohydrates, the easier it is to produce polycyclic aromatic hydrocarbons”:

3)     In the same sense, I have some doubts about the conclusions related to the addition of basic and/or acid amino acids to sausages (i.e. “it is interesting to note that the addition of basic amino acids is capable of producing more PAHs than adding acidic amino acids”), since the evaluation of only two basic and two acid amino acids does not seem enough to support such statement.

4)     In general, there are several editing and writing errors. English should also be improved. Furthermore, authors should be pay attention to the presentation of their manuscript when submitting it to any journal (e.g. lines are numbered from section 2.3 and not before, use of capital letters when it is not needed (line 33,…found that The pyrolysis…), author’s affiliations are missing, etc.). A poor presentation of the manuscript could be reason enough for its rejection. In addition, Tables 2 and 3 should be edited in order to clarify the results.

Specific comments:

Comment 1 (line 51): The results achieved from the evaluation of L-valine as food additive are not shown either in Figure 2 or Table 3. These results should be included in the manuscript and discussed.

Comment 2: Table 3 has not been referred in the main text.

Comment 3: Several of the experiments are based on the differences in the molecular weight and polarity of food additives, but the characteristics of the studied compounds are not indicated in the manuscript. A table indicating the molecular weight, pka and/or pkb, and logP of each studied compound is required.

Comment 4: (lines 167-168). The terms “R groups and R-based” are continuously used by the authors. However, the meaning of both terms is not clear. This issue should be clarified.

Comment 5: Based on the introduction, it seems that authors are going to evaluate the formation of PAHs as a consequence of a grilling process. However, in their experiments, sausages are finally baked. Authors should be aware that this could be quite confusing for the reader. Moreover, the formation of PAHs can be different based on the cooking processes. This should be taken into account when comparing their results with the results achieved by other authors.

Comment 6: In section 2.1., LODs and LOQs are expressed as µg/kg but both terms are expressed as ng/mL in Table 1. Since method validation was carried out using standard solutions, concentrations should be expressed as ng/mL. In this sense, the influence of the matrix on the analytical response should be evaluated in order to discard any matrix effect and to confirm that the use of standard calibration curves for the quantification of PAHs in sausages is possible.

Comment 7 (Table 3 caption): The meaning of “a, b, c, d, e, indicate significant…” is not clear.

Comment 8 (Section 3.4. Extraction and clean-up): What kind of shaking process was used for the extraction of analytes? During SPE, were the columns submitted to any washing step before the elution of the PAHs?

Author Response

Response to Reviewer 1 Comments

Thank you so much for your comments and suggestions. Our manuscript was revised based on your comments. The details were shown as following.

Point 1: Only one keto-based carbohydrate (i.e. D-fructose) and one aldehyde-based carbohydrate (D-glucose) are evaluated. The sample size is not representative for concluding that “R-based aldehyde-based carbohydrates are capable of producing more PAHs than R-based keto-based carbohydrates during high-temperature processing of foods”. The formation of PAHs produced as a consequence of other carbohydrates of each type must be studied in order to support this conclusion. 

Response 1: Thank you so much for your correction. D-glucose and D-fructose are a pair of isomers which have the same molecular weight, except that the R group of D-glucose is an aldehyde group (CH2OH-(CHOH)4-CHO), and the R group of D-fructose is a keto group (CH2OH-(CHOH)3-(C=O)-CH2OH). So the author has reason to imagine that the effect of adding D-glucose and D-fructose on the PAHs content in grilled pork sausages is caused by these two different R groups. Of course, the authors also realize that this result does not seem to be enough to draw the conclusion of "R-based aldehyde-based carbohydrates are capable of producing more PAHs than R-based keto-based carbohydrates during high-temperature processing of foods". We have changed it to "The results showed that the grilled sausage with the aldehyde-based D-glucose was capable of producing more PAHs than the sausage with the keto-based D-fructose" (See line 32-33 for details.); The results showed that addition of aldehyde-based D-glucose could significantly increase the PAHs production when compared with that of the keto-based D-fructose group.’’(See line 303-305 for details.)

Point 2: Furthermore, only three carbohydrates were selected for the evaluation of their molecular weight on the formation of PAHs during food processing. The sample size is not enough to conclude that “the smaller the molecular weight of carbohydrates, the easier it is to produce polycyclic aromatic hydrocarbons”

Response 2: 4-(α-D-Glucosido)-D-glucose consists of two molecules of glucose, Molecular weight: 360.32; cellulose is generally composed of 8,000 to 10,000 glucose residues linked by β-1,4-glycosidic bonds. Molecular weight: (162.06) n, molecular weight usually >50000. The authors have reason to believe that the effect of adding these three carbohydrates on the PAHs content in grilling pork sausages is due to the difference in molecular weight. But the author has revised the conclusion to: "A higher PAHs content was determined in the the grilled sausage with smaller molecular weight D-glucose when compared with the larger molecular weight 4-(α-D-Glucosido)-D-glucose and cellulose. However, there was no significant difference in the content of PAHs in grilled pork sausages with the addition of small molecular weight D-fructose and large molecular weight cellulose (P > 0.05). This result may be because the effect of carbohydrates on PAHs production was affected by both R group and molecular weight, and the effect of R group on PAHs formation was greater than the effect of molecular weight. "See line 139-145 for details.

Point 3: In the same sense, I have some doubts about the conclusions related to the addition of basic and/or acid amino acids to sausages (i.e. “it is interesting to note that the addition of basic amino acids is capable of producing more PAHs than adding acidic amino acids”), since the evaluation of only two basic and two acid amino acids does not seem enough to support such statement.

Response 3: Thank you so much for your correction. The authors selected some amino acids commonly used in the processing of meat products as research objects. However, it is not appropriate to select only two acidic and two basic amino acids to give the result "it is interesting to note that the addition of basic amino acids is capable of producing more PAHs than adding acidic amino acids". The author has revised the description of this result. For example, "In addition, when comparing the effects of acidic amino acids (L-lysine and L-arginine) and basic amino acids (L-glutamic acid and L-aspartate) on the formation of PAHs in grilled pork sausages, the authors found that the content of PAHs in grilled pork sausages with basic amino acids (L-glutamic acid and L-aspartate) was significantly higher than that in grilled pork sausages with acidic amino acids (L-lysine and L-arginine) (P<0.05) (details are shown in Table 3)." The author has clearly pointed out in the article that the two basic amino acids, L-lysine and L-arginine, are more likely to form polycyclic aromatic hydrocarbons than the two acidic amino acids, L-glutamic acid and L-aspartate. See line 187-191,309-310 for details.

Point 4: In general, there are several editing and writing errors. English should also be improved. Furthermore, authors should be pay attention to the presentation of their manuscript when submitting it to any journal (e.g. lines are numbered from section 2.3 and not before, use of capital letters when it is not needed (line 33,…found that The pyrolysis…), author’s affiliations are missing, etc.). A poor presentation of the manuscript could be reason enough for its rejection. In addition, Tables 2 and 3 should be edited in order to clarify the results.

Response 4: Thank you so much for your correction. The author has reviewed the English spelling of the manuscript and made corresponding corrections, and the line number of the manuscript has been revised again. Table 2 and Table 3 have also been re-edited.

Point 5: Comment 1 (line 51): The results achieved from the evaluation of L-valine as food additive are not shown either in Figure 2 or Table 3. These results should be included in the manuscript and discussed.

Response 5: Thank you so much for your correction. The authors wanted to compare the differences in PAHs production between L-tryptophan, L-phenylalanine and L-proline. The appearance of L-valine is caused by the author's spelling mistakes, so I am very sorry for the doubts. The author has reviewed and revised the statement as follows: "It was found that the content of PAHs in grilled pork sausages supplemented with L-tryptophan and L-phenylalanine was significant lower than sausage with L-proline." See line 210-211 for details.

Point 6:  Table 3 has not been referred in the main text.

Response 6: Thanks for your suggestion. The author has already marked Table 3 in the manuscript. See line 175,177 for details.

Point 7:  Several of the experiments are based on the differences in the molecular weight and polarity of food additives, but the characteristics of the studied compounds are not indicated in the manuscript. A table indicating the molecular weight, pka and/or pkb, and logP of each studied compound is required.

Response 7: Thanks for your suggestion. The authors have added information on carbohydrate molecular weight and amino acid characteristics as required, as follows: D-glucose (molecular weight:180.16) (CAS number: 50-99-7), D-fructose (molecular weight:180.16) (CAS number: 53188-23-1), 4-(α-D-Glucosido)-D-glucose (molecular weight:360.32) (CAS number: 6363-53-7), cellulose (molecular weight:(162.06)n) (CAS number: 9004-34- 6), Non-polar amino acids: L-proline (CAS number: 147-85-3), L-tryptophan (CAS number: 73-22-3) and L-phenylalanine (CAS number: 63-91-2 ); Polar amino acids: L-tyrosine (CAS number: 60-18-4), L-threonine (CAS number: 72-19-5) and L-serine (CAS number: 56-45-1); Basic amino acids: L- Lysine (CAS number: 56-87-1) and L-arginine (CAS number: 74-79-3); Acidic amino acids: L-glutamic acid (CAS number: 56-86-0) and L-aspartic Amino acid (CAS No: 56-84-8). See line 227-234 for details.

Point 8:  Comment 4: (lines 167-168). The terms “R groups and R-based” are continuously used by the authors. However, the meaning of both terms is not clear. This issue should be clarified.

Response 8: Thank you so much for your correction. R groups (the R group represents an aldehyde group in a glucose molecule and a ketone group in a fructose molecule). The author has clarified the meaning of the R base in the manuscript as required. See line 28-29 for details.

Point 9:  Comment 5: Based on the introduction, it seems that authors are going to evaluate the formation of PAHs as a consequence of a grilling process. However, in their experiments, sausages are finally baked. Authors should be aware that this could be quite confusing for the reader. Moreover, the formation of PAHs can be different based on the cooking processes. This should be taken into account when comparing their results with the results achieved by other authors.

Response 9: I am very sorry for the confusion caused by my misrepresentation. In fact, all the sausages in this study were processed using “Grilled”. The author has thoroughly reviewed the manuscript and revised it. 

Point 10:  Comment 6: In section 2.1., LODs and LOQs are expressed as µg/kg but both terms are expressed as ng/mL in Table 1. Since method validation was carried out using standard solutions, concentrations should be expressed as ng/mL. In this sense, the influence of the matrix on the analytical response should be evaluated in order to discard any matrix effect and to confirm that the use of standard calibration curves for the quantification of PAHs in sausages is possible.

Response 10: First of all, thank you very much for your advice. However, the authors think that LODs and LOQs are suitable to express as μg/kg. In fact, many scholars have used similar expressions. For example: "Table 2 shows the linearity equations, with R2 values ranged from 0.993 to 0.999 for all the three PAHs standards. Table 2 shows LOD of 0.02, 0.01 and 0.03 μg/kg and LOQ of 0.09, 0.04 and 0.10 for F, BbF and BaP, respectively." (Farhadian, A., Jinapac, S., & Zaidul, I. S. M. (2012). Effects of marinating on the formation of polycyclic aromatic hydrocarbons (benzo[a]pyrene, benzo[b]fluoranthene and fluoranthene, Food Control, 28(2), 420-425.) in grilled beef meat. The PAHs were quantified using an external standard method. The standard curve of each PAHs was prepared by plotting the peak area against standard concentration (ng/ml), and the amount of each PAH was calculated on the basis of its respective calibration curve. The limits of detection (LOD) and quantitation (LOQ) were calculated by using the signal-to-noise ratio of S/N =3 and S/N = 10,respectively. The average LOD of 16 PAHs ranged from 0.005 to 0.36 μg/kg.”(Li, G., Wu, S., Wang, L., & Akoh, C. C. (2015). Concentration, dietary exposure and health risk estimation of polycyclic aromatic hydrocarbons (pahs) in youtiao, a chinese traditional fried food. Food Control, 59, 328-336.)

Point 11:  Comment 7 (Table 3 caption): The meaning of “a, b, c, d, e, indicate significant…” is not clear.

Response 11: The author has modified the manuscript as required. See line 201 for details.

Point 12: Comment 8 (Section 3.4. Extraction and clean-up): What kind of shaking process was used for the extraction of analytes? During SPE, were the columns submitted to any washing step before the elution of the PAHs?

Response 12: After shaking for 5 minutes with a quick mixer (SK-1, Jintan is Jairel Electric Co., Ltd., China); During the SPE, the column did not undergo any washing steps prior to elution of the PAHs.

Reviewer 2 Report

The manuscript entitled "Small molecular weight aldose and basic amino

acids increase the occurrence of PAHs in roasted

pork sausages" by Nie et al. reports a study focused on effect of type of 

aminoacids and carbohydrates on formation of PAH in meat. The topic is interesting and 

relevant, since knowledge of impact of different additiveson formation of PAH has practical value.

Differences in formation of PAHs from different aminoacids are already known:

(please compare e.g.: Sharma et al. https://doi.org/10.1016/j.jaap.2005.03.010). 

It may be expeced, that their addition to a food product will have similar effect.

Also some interactions between presence of carbohydrates and PAHs formation are known:

(Britt et al. https://doi.org/10.1016/j.fuel.2004.02.009; GOng et al. https://doi.org/10.1016/j.supflu.2016.03.021). 

So the authors should more clearly indicate in the introduction what exactly was done previously in 

this matter and what is the real novelty of the presented research. 

Some parts of manuscript need detailed writing/ language correction, below just some of the examplesL

- abstract: 

line 1 "wildly"

line 2 "proved be" 

line 4 " was still few known"

line 4: R groups - what are R groups?

-line 35 "..to form Amalido[29]. The Amalino.."

- chapters 3.1, 3.5. , 3.6 - numerous missing spaces, 

- line 49 ".. it was found that the content of PAHs in roasted pork sausages supplemented with L-49 tryptophan and L-phenylalanine was significantly lower than that of added."

- line 54 "Ramesh K. Sharma and W. Geoffrey Chan et al. found that aspartic acid and proline degradable at high temperatures to produce polycyclic aromatic hydrocarbons ..."

Manuscript is missing:

- affiliations

- page and line numbering in half of the manuscript

Other comments:

- tab 3 - means of how many determinations

- some of the citations in text are incorrect, eg. contain first name of authors eg. (non numbered page): "Thomas, E studied.... ";  line 54 "Ramesh K. Sharma and W. Geoffrey Chan et al.", 

- please check reference style, eg. reference 15 "[15]Gemma Perelló, Roser Martí-Cid, Castell, V. , Llobet, J. M. , & José L Domingo. (2009). ..." 

reference 27 " [27]Thomas E, McGrath, W.Geoffrey ChanLow temperature mechanism for the formation of polycyclic aromatic 221 hydrocarbons from the pyrolysis of cellulose[J]"  ecc. 

- some conclusions are not in accordance with the obtained data: 

for example line 143: "The molecular weight of carbohydrates is also a factor influencing PAHs. Smaller molecular weight carbohydrates produce more PAHs than larger molecular weight carbohydrates." 

while total amount of PAH in table 2 does not significantly differ for fructose, and cellulose

Author Response

Response to Reviewer 2 Comments

Thank you so much for your comments and suggestions. Our manuscript was revised based on your comments. The details were shown as following.

Point 1: Differences in formation of PAHs from different amino acids are already known: (please compare e.g.:Sharma etal. https://doi.org/10.1016/j.jaap.2005.03.010). It may be expeced, that their addition to a food product will have similar effect.  Also some interactions between presence of carbohydrates and PAHs formation are known:  (Britt et al. https://doi.org/10.1016/j.fuel.2004.02.009; GOng et al. https://doi.org/10.1016/j.supflu.2016.03.021).   So the authors should more clearly indicate in the introduction what exactly was done previously in   this matter and what is the real novelty of the presented research.

Response 1: Thanks for your suggestion. In the preface, the previous studies were reviewed and the differences between my research and the predecessors were marked. See line 63-75 for details.

Point 2:Some parts of manuscript need detailed writing/ language correction, below just some of the examples“abstract:   line 1 "wildly"  line 2 "proved be"   line 4 " was still few known"  line 4: R groups - what are R groups?    -line 35 "..to form Amalido[29]. The Amalino.." - chapters 3.1, 3.5. , 3.6 - numerous missing spaces,”

Response 2: Thanks for your suggestion. The author has reviewed and revised all manuscripts in detail and added information such as the author's affiliates. The author re-edited the manuscript line number and added the missing part.

Point 3: tab 3 - means of how many determinations

Response 3: Table 2 and Table 3 are all average values determined by three measurements. See line 156,199 for details.

Point 4:- please check reference style, eg. reference 15 "[15]Gemma Perelló, Roser Martí-Cid, Castell, V. , Llobet, J. M. , & José L Domingo. (2009). ..."   reference 27 " [27]Thomas E, McGrath, W.Geoffrey ChanLow temperature mechanism for the formation of polycyclic aromatic 221 hydrocarbons from the pyrolysis of cellulose[J]"  ecc.

Response 4: The author has conducted a detailed review of all references in the manuscript and made corrections. See line 358,363 for details.

Point 5: some of the citations in text are incorrect, eg. contain first name of authors eg. (non numbered page): "Thomas, E studied.... ";  line 54 "Ramesh K. Sharma and W. Geoffrey Chan et al.",

Response 5: The author has revised some of the citations in text.

Point 6:- some conclusions are not in accordance with the obtained data:   for example line 143: "The molecular weight of carbohydrates is also a factor influencing PAHs. Smaller molecular weight carbohydrates produce more PAHs than larger molecular weight carbohydrates."while total amount of PAH in table 2 does not significantly differ for fructose, and cellulose

Response 6: The author has revised the manuscript, and the effect of the molecular weight of carbohydrates on the formation of PAHs is revised to:“Smaller molecular weight D-glucose produce more PAHs than larger molecular weight 4-(α-D-Glucosido)-D-glucose and cellulose. However, there was no significant difference in the content of PAHs in grilling pork sausages with the addition of small molecular weight fructose and large molecular weight cellulose (P > 0.05).” See line 139-145, 306-308 for details.

Reviewer 3 Report

The article is well written and it deals with a very important subject that is the ingestion of carcinogenic substances with food. The experiments are conducted correctly and conclusions are supported by results.

1. The authors should describe which PAHs have been selected and why.

2. Effect of carbohydrate: add a figure analogue to figure 2 to illustrate results

Author Response

Response to Reviewer 3 Comments

Thank you very much for your confirmation of my research! Thank you also for your suggestions. Our manuscript was revised based on your comments. The details were shown as following.

Point 1: The authors should describe which PAHs have been selected and why.

Response 1: The author has already written in the manuscript to study "Twelve kinds of PAHs were used as standards (Naphthalene (NA), Acenaphthene (Ac), Fluorene (FLU), Fluoranthene (FLT), Benzo [a] anthracene (BaA), Chrysene ( CHR), Benzo[b]fluoranthene (BbFA), Benzo[k]fluoranthene (BkFA),Benzo[a]pyrene (BaP), Dibenzo[a,h]anthracene (DBahA),Benzo[g,h,i]perylene (BPE), Indeno[1,2,3-c,d]pyrene (IPY)" (See line 237-242 for details.). At the same time, the author indicates the reason " In this paper, we tested the effects of the property of carbohydrates with different R-base (an aldehyde group in a glucose molecule and a ketone group in a fructose molecule) and different molecular weight, and the property of amino acids (with different polarities and different acid-base) on the formation of twelve kinds of PAHs in grilled sausage. All of these PAHs were noted as potential food contaminants by the United States Environmental Protection Agency [19]." (See line 88-93 for details.)

Point 2: Effect of carbohydrate: add a figure analogue to figure 2 to illustrate results

Response 2: The author believes that the description of the information in Table 2 through the histogram is a repeated description. The author believes that the form can convey information to the reader more accurately.

Round 2

Reviewer 1 Report

Authors have answer to reviewers' comments and have improved the quality of their manuscript. However, the following issues must be solved before its publication.

In relation to:

- "Response 7: Thanks for your suggestion. The authors have added information on carbohydrate molecular weight and amino acid characteristics as required, as follows..."

Comment 7.2: The table containing the following information "the molecular weight, pka and/or pkb, and logP of each studied compound" is still required.

- "Response 10First of all, thank you very much for your advice. However, the authors think that LODs and LOQs are suitable to express as μg/kg. In fact, many scholars have used similar expressions..."

I do not deny that external standard calibration curves can be used for compound quantification in samples but, if method characterization is carried out using standard solutions, solutions concentration is referred to weight/volume. Obviously, when applying to solid samples, concentration must be expressed in weight/weight, and the application of a dilution/pre-concentration factor could also be required. Regarding LOQs/LODs, if they are calculated using standard solutions, these values could not be applicable to samples since sample dilution or pre-concentration can result from sample treatment. In these case, new LOQs/LODs values expressed in weight/weight must be estimated.

- "Response 11The author has modified the manuscript as required. See line 201 for details."

From my point of view, it is still not clear what " significant differences between the different rows" since the rows have not been listed. 

Other minor corrections:

Line 24. "widely" instead of "wildly"

Line 24. "aditives" instead of "aditive"

Line 25. "...have proven to have an influence on the production..."

Line 34. "...D-glucose compared to the sausage..."

Table 2 Figure caption. "Concentrations (n = 3) of 12 PAHs produced as a consequence of adding different carbohydrates to grilled sausages" (Idem Table 3 Figure caption).

Author Response

Response to Reviewer 1 Comments

Thank you so much for your comments and suggestions. Our manuscript was revised based on your comments. The details were shown as following.

Point 1: Comment 7.2: The table containing the following information "the molecular weight, pka and/or pkb, and logP of each studied compound" is still required. 

Response 1: Thank you so much for your suggestion. The authors have added the molecular weight, PKa and logP of each study compound according to your recommendations.See line Table 4 for details.

Point 2: From my point of view, it is still not clear what " significant differences between the different rows" since the rows have not been listed.

Response 2:I am very sorry for the inaccuracy in my expression. I have made a revision, such as "a, b, c, d, e, different letters indicate significant differences in PAHs content between any two rows in Table 3 (P < 0.05)"

Point 3: Other minor corrections:    Line 24. "widely" instead of "wildly"  Line 24. "aditives" instead of "aditive"  Line 25. "...have proven to have an influence on the production..."  Line 34. "...D-glucose compared to the sausage..."  Table 2 Figure caption. "Concentrations (n = 3) of 12 PAHs produced as a consequence of adding different carbohydrates to grilled sausages" (Idem Table 3 Figure caption).

Response 3: Thank you so much for your correction. The author has carefully reviewed the manuscript again and revised the language errors and marked them in red in the manuscript. Thank you very much for your suggestion for Table 2 Figure caption, which has been modified by the author.

Reviewer 2 Report

The manuscript was improved but it needs still very detailed and careful correction e.g. of language, reference style. For example:

- line 49 contribut -> contribute

- line 58-59 "Grilling is a cooking method that using the high temperature of the heating source
to destroy pathogenic and microorganisms, and to make the food taste good."  

- line 194 "Amalido compound"??? did you mean Amadori compound?

Please again very carefully check the reference style, and correctness.

e.g.:
- line 330: "[3]Authority, E. F. S. (2007). Findings of the efsa data collection on polycyclic aromatic hydrocarbons in food. E330 fsa Journal, 5(9),3-55."

 it is not correct way of citing the  European Food Safety Authority document.

- line 355, 358, 391 etc. - given names in full shouldn't be included in the references

Author Response

Response to Reviewer 2 Comments

Thank you so much for your comments and suggestions. Our manuscript was revised based on your comments. The details were shown as following.

Point 1: line 49 contribut -> contribute  - line 58-59 "Grilling is a cooking method that using the high temperature of the heating source  to destroy pathogenic and microorganisms, and to make the food taste good."    - line 194 "Amalido compound"??? did you mean Amadori compound?

Response 1: Thanks for your suggestion. Thank you so much for your correction. The author has carefully reviewed the manuscript again and revised the language errors and marked them in red in the manuscript. - line 194 "Amalido compound" refers to 1-amino-1-deoxy-2-ketosaccharide, which was first discovered by Amalido, so it is also called "Amalido compound". The author has already defined the meaning of "Amalido compound" in the text. See line 195 for details.

Point 2:Please again very carefully check the reference style, and correctness.  e.g.:  - line 330: "[3]Authority, E. F. S. (2007). Findings of the efsa data collection on polycyclic aromatic hydrocarbons in food. E330 fsa Journal, 5(9),3-55."   it is not correct way of citing the  European Food Safety Authority document.  - line 355, 358, 391 etc. - given names in full shouldn't be included in the references

Response 2: Thanks for your suggestion. The author has reviewed and revised all manuscripts in detail, and the author modified the format of the reference as required by the magazine.See line 329-405 for details.
